# Effect of Resonant Frequency Vibration on Delayed Onset Muscle Soreness and Resulting Stiffness as Measured by Shear-Wave Elastography

**DOI:** 10.3390/ijerph18157853

**Published:** 2021-07-24

**Authors:** Garrett C. Jones, Jonathan D. Blotter, Cameron D. Smallwood, Dennis L. Eggett, Darryl J. Cochrane, J. Brent Feland

**Affiliations:** 1Department of Mechanical Engineering, College of Engineering, Campus of Brigham Young University, Provo, UT 4019, USA; garrettjones.co@gmail.com (G.C.J.); jblotter@byu.edu (J.D.B.); cameronsmallwood01@gmail.com (C.D.S.); 2Department of Statistics, College of Physical and Mathematical Sciences, Brigham Young University, Provo, UT 4019, USA; theegg@byu.edu; 3School of Sport, Exercise & Nutrition, College of Health, Massey University, Palmerston North 4442, New Zealand; d.cochrane@massey.ac.nz; 4Department of Exercise Sciences, College of Life Sciences, Campus of Brigham Young University, Provo, UT 4019, USA

**Keywords:** vibration, whole-body, skeletal muscle, shear-wave, elastography, eccentric, delayed onset muscle soreness, recovery, biceps

## Abstract

This study utilized resonant frequency vibration to the upper body to determine changes in pain, stiffness and isometric strength of the biceps brachii after eccentric damage. Thirty-one participants without recent resistance training were randomized into three groups: a Control (C) group and two eccentric exercise groups (No vibration (NV) and Vibration (V)). After muscle damage, participants in the V group received upper body vibration (UBV) therapy for 5 min on days 1–4. All participants completed a visual analog scale (VAS), maximum voluntary isometric contraction (MVIC), and shear wave elastography (SWE) of the bicep at baseline (pre-exercise), 24 h, 48 h, and 1-week post exercise. There was a significant difference between V and NV at 24 h for VAS (*p* = 0.0051), at 24 h and 1-week for MVIC (*p* = 0.0017 and *p* = 0.0016, respectively). There was a significant decrease in SWE for the V group from 24–48 h (*p* = 0.0003), while there was no significant change in the NV group (*p* = 0.9341). The use of UBV resonant vibration decreased MVIC decrement and reduced VAS pain ratings at 24 h post eccentric damage. SWE was strongly negatively correlated with MVIC and may function as a predictor of intrinsic muscle state in the time course of recovery of the biceps brachii.

## 1. Introduction

Exercise-induced muscle damage occurs after an individual performs unaccustomed exercise and results in various muscle damage symptoms such as delayed-onset-muscle-soreness (DOMS) [1], loss of muscle strength with reduced electromyography (EMG) activity [2], and muscle stiffness [1]. Eccentric muscle damage has been shown to include extracellular matrix and myofibrillar disruption [3], inflammatory cell infiltration of damaged fibers [4], disruption of z-lines [5], and increased satellite cell activation [6]. Symptoms of DOMS can include pain (which can often be severe), swelling, and reduced joint range of motion (ROM) [7]. The measurement of either concentric or isometric strength is considered to be an important, reliable and valid indicator of the extent of muscle damage [8]. The combination of strength loss and pain can alter muscle recruitment patterns and increase risk of injury [9]. Research into clinically applicable adjunctive methods to aid muscle recovery from DOMS (such as ice or cold water immersion [10,11] and massage [12]) has been previously conducted with varied results, but have been reported to primarily mitigate muscle soreness [7], while a meta-analysis on the use of compression garments shows some effect on the return of muscle power [13]. Despite a large number of studies on recovery strategies for DOMS, the actual mechanisms and underpinnings of DOMS remain unclear [7,14].

Recently, the whole-body vibration (WBV) platform has been introduced as a potential method to decrease the symptoms of DOMS [15,16,17]. WBV has been reported to improve blood flow velocity [18], microvascular circulation in the muscle [19], muscle perfusion [20], and facilitates muscle temperature increases [21], making it an important modality to consider for aiding muscle recovery. Currently, there are a limited number of WBV platform DOMS intervention studies that have either shown no difference compared to an exercising control group [22,23] or have reported improvements in indirect pain measures [16,17,24], with one study reporting attenuation of strength loss [17]. In previous WBV studies, the exercise induced muscle damage (EIMD) protocols and the application of vibration. Prior to [16,17] or after [22,23,24], the muscle damage protocol has varied. Similarly, in earlier research, vibration parameters have varied. For example, vibration frequency and amplitude have ranged from 20–50 Hz and 2–4 mm, respectively, and duration has ranged from 2 bouts of 30-s to 10 min total. Additionally, the use of WBV platform DOMS studies have been limited in targeting the quadriceps muscle group, with previous vibration protocols employing a static half squat or dynamic squat. However, little is known about the efficacy of using the WBV platform in upper extremity DOMS and requires further investigation.

Shear-wave elastography has recently emerged as a promising assessment tool with the potential for diagnosing and monitoring muscle changes [25] and as a noninvasive method to estimate muscle damage resulting from damaging exercise [26]. To date, the efficacy of WBV as a DOMS intervention strategy has been limited to common indirect measures of EIMD: isometric force [16,17,23], visual analog pain scales [16,17,22,23,24], pressure pain threshold [16,17,22], limb swelling [17,22], loss of range of motion or flexibility [22,23,24], and creatine kinase levels [17]. Recent studies have shown that shear-wave elastography (SWE) can be used as a measure of muscle stiffness [27,28]. Shear-wave elastography differs from strain elastography in that it can provide a quantitative measure of intrinsic muscle stiffness calculated from the shear-wave velocity [29]. The shear modulus from SWE has been shown to be strongly correlated with Young’s modulus when the ultrasound transducer is parallel to the muscle fiber arrangement of the muscle being measured [30]. While SWE has shown promise in studying muscle changes with age [28], muscular disorders such as Duchenne’s [31], and idiopathic inflammatory myopathies [32], its efficacy as a potential measurement tool for DOMS is scant and warrants investigation. Furthermore, the relationship between stiffness and strength loss resulting from EIMD is unclear and warrants investigation.

Thus, the purpose of this study was to determine the effect of using a WBV platform for upper body vibration (UBV) on pain, isometric strength, and muscle stiffness recovery of the bicep brachii muscle following a muscle damage protocol as evaluated by SWE.

We hypothesized that UBV would improve symptoms of exercise-induced muscle damage by reducing subjective ratings of pain, improving isometric strength, and decreasing muscle stiffness.

## 2. Materials and Methods

### 2.1. Experimental Design

A randomized single-blinded trial was implemented to investigate the effect of UBV on ratings of perceived pain, muscle strength, and muscle stiffness. Each participant was randomly assigned to one of three groups by drawing a number from a bowl: Control (C), and two exercise groups (No vibration (NV) and Vibration (V)). An EIMD protocol was designed to produce muscle damage in this study. Muscle damage was defined as the presence of MVIC force loss and was expected to be between 30–50% after the EIMD protocol (post-ex). The EIMD protocol was performed by groups NV and V following baseline measurements of pain (VAS), maximal voluntary isometric contraction (MVIC), and SWE of the biceps brachii. Participants in the V group received vibration therapy exercise to the upper extremities for four out of seven days following the EIMD protocol. Group NV did not receive any vibration therapy at any time after the exercise protocol. The C group did not perform the EIMD protocol or undergo any UBV, but completed the same assessments as the treatment groups. The use of a true control group (where participants did not participate in any training program) is common in vibration related literature [33] and was used to adequately account for variance and ensure the reliability of our dependent measures when assessing both EIMD or non-EIMD responses. The independent variables were the three groups and the dependent variables for all groups were SWE, MVIC, and VAS measurements performed at five time intervals: (1) Baseline (prior to any interventions), (2) post exercise (post-ex), (3) 24 h post exercise (24 h), (4) 48 h post exercise (48 h), and (5) 1-week post exercise (1-week). The only exception was SWE, which was not measured at post-ex. A visual analog scale (VAS) was used as a subjective measure of pain in response to the muscle damage protocol. A maximum voluntary isometric strength test (MVIC) indirectly assessed muscle damage. Ultrasonic SWE represented a measure of intrinsic muscle stiffness. The reporting of this study conforms to the STROBE (strengthening the reporting of observational studies in epidemiology) guidelines [34].

### 2.2. Participants

Thirty-one participants completed this study (28 males, 3 females: age 23.8 ± 2.7 years.; height 177.4 ± 7.7 cm; weight 74.1 ± 10.5 kg). Final participant numbers per group were: C (9 males, 2 females: age 23.3 ± 2.4 years.; height 175.5 ± 8.1 cm; weight 73.9 ± 12.3 kg); NV (10 males: age 23.7 ± 3.2 years.; height 175.2 ± 6.3 cm; weight 70.1 ± 8.0 kg); V (9 males, 1 female: age 24.5 ± 2.7 years.; height 181.6 ± 7.7 cm; weight 78.4 ± 10.5 kg). The sample size was analyzed using the G*Power program version 3.1.9.5 (Franz Faul, University of Kiel, Germany). With an estimated effect size of 0.71 and power at 0.80, 26 participants were required for statistical significance. To allow for noncompliance or drop-out, 34 participants were recruited. From this, three participants did not complete the study (one withdrew from pain and two withdrew due to incomplete measurements). The inclusion criteria for this study required participants to be 18 years or older, recreationally active (defined for this study as exercising 3 x/week for 30 min or more) and not have participated in upper-body resistance training for the past 3 months. Participants were also required to have no current or recent history of noticeable musculoskeletal joint pain or muscle pain related to DOMS, acute joint disease or history of arm or shoulder injury in the past six months. Participants were instructed not to participate in any strenuous activity 48 h before the study and to avoid new or strenuous activity that required the use of their arms during the 1-week research period. Participants were also asked to avoid use of other adjunctive recovery therapies during the experiment such as, ice, massage, topical ointments, or medications.

All participants were informed of the possible risks associated with the research and signed a university approved consent form. Approval for this study was received from the University’s Institutional Review Board (IRB) (IRB #18077).

### 2.3. Maximal Isometric Strength and EIMD Protocol

Upon reporting to the laboratory, each participant had their initial VAS and SWE (baseline) measurements recorded, followed by MVIC. MVIC was measured using a 113 kg strain gauge load cell (Sentran, Ontario, CA, USA) fixed via a swivel connector to a base plate attached to the floor. This measurement was performed by the participant contracting isometrically on a cable with their right arm at a 125° elbow flexion angle. To prevent unwanted movement participants were instructed to stand with heels, back, elbow, and head against the wall. If shoulder shrugging or other compensatory trunk movements were identified, the participant repeated the measurement. Participants performed three repetitions for three seconds and the maximum value from the three attempts was recorded. Participants in both the V and NV groups performed the EIMD protocol by completing 100 total repetitions of a dumbbell arm curl exercise (10 sets × 10 repetitions). Each repetition was performed at a cadence of approximately one-second concentrically raising the arm and three-seconds eccentrically lowering by manual counting. The first two sets were completed using 50% of each participant’s MVIC measurement, and the remaining sets (3–10) were completed with the load reduced by 2.2 kg. One-minute rests were enforced between each set. Immediately following the damage protocol, MVIC, and VAS were re-evaluated (post-ex). Group V then received UBV therapy while participants in the NV group did not receive any intervention. Participants in the control group (C) did not perform the EIMD protocol and had their baseline measurements rerecorded.

### 2.4. Muscle Soreness

Muscle soreness was measured using a self-reported 100-mm long visual analog scale with “no pain at all” (0 mm) and “worst pain imaginable” (100 mm). The participant rated their current pain by moving their right arm through their full range of motion from full elbow extension to full elbow flexion and then to extension. Participants then rated their perceived soreness by placing a single vertical line through the VAS, which has been shown to be a valid and reliable measure of pain [35]. The distance (in mm) from no pain to the mark indicated was measured and converted to cm for data analysis.

### 2.5. Shear Wave Elastography (Stiffness)

All SWE measurements were made using an ultrasound transducer (GE Logiq S8 and a 9 L head (GE Healthcare, Chicago, IL, USA). Participants were evaluated for “stiffness” in the lower half of the bicep muscle via ultrasound SWE. All SWE measurements were recorded by a trained physical therapist who was blinded to treatment and had 4 years of experience with SWE measurements. The lower half of the biceps was selected based on our experience that the locality of soreness normally occurs at the distal end of muscles after EIMD. Participants lay supine on a treatment table with the arm relaxed and extended at their side. Previously, it has been noted that lengthened muscle positions increase the SWE values and muscle contraction [29]. From pilot work we found that SWE intrarater reliability of the biceps brachii was improved in a relaxed and extended position compared to the resting position (approximately 90° elbow flexion), with intraclass coefficients of 0.90 vs. 0.74, respectively. However, this may have been a function of the seated positioning of the participants and ease of access and probe positioning by the ultrasonographer. Contrarily, Dubois et al. [36] reported that measurement reliability of lower limb muscles was not affected by position. Importantly, in the current study the arm was not fully extended or fully supinated position, but rather a small towel supported the lower wrist and hand. The positioning of the ultrasound probe was parallel with the longitudinal axis of the bicep muscle belly. This is important since SWE has been shown to be a valid estimator of stiffness in this position by accurately modeling Young’s modulus when compared to standard materials testing [30]. To improve reliability of positioning the ultrasound head for subsequent tests, a permanent pen marked the borders of the ultrasound head. Participants were instructed to maintain markings and if fading occurred the exact sites were remarked during subsequent visits. An elastograph box (1–1.25 cm^2^) was placed within the confines of the bicep muscle at a depth between 1–3 cm, starting as close to 1 cm as possible depending on vasculature and fascial lines. Variance in the elastogram increases as depth increases and elastograph images should be measured at least 1 cm from the probe [29,37]. Care was taken to place the box in a position with no apparent vasculature and avoid larger fascial lines as these affect the SWE results [29]. Once the appropriate probe pressure and position was achieved, the elastogram recording continued until it maintained a consistent colorization for 3–4 s. The stiffness rating was recorded from four separate screen shot samples, which were taken from the cineloop recording. While other studies have used various size regions of interest (ROI) circles placed throughout the elastogram to calculate stiffness [38], we have observed less variation in SWE values using an elastogram area calculation. There are many SWE systems available with different processing and algorithms to quantify elasticity, and there is limited research comparing reproducibility between systems as well as methods of ROI or elastogram processing [39]. From our pilot work, the ROI placement method resulted in a range of 2.5–10% variation in repeated measurements of a particular elastogram (with smaller ROI circles creating more variation than larger circles) as compared to 0.5–1.5% variation using an area calculation for the biceps brachii (which is longitudinal). To date, we have not piloted assessment in pennated muscles but surmise medium size ROI’s will be more consistent as reported in a recent study [39]. Thus, SWE for our study was calculated using a stiffness area calculation by outline tracing of the elastogram and the ultrasound transducer software provided the stiffness value in kilopascals (kPa). The overall stiffness value used for analysis was the average of the four samples taken from the cineloop recording.

An example of SWE results from pre to 24 hr is shown in Figure 1. The elastogram shows colors in the samples relative to stiffness. For the current settings, the scale ranged from 0–150, with 0 and 150 indicating red and dark blue, respectively.

### 2.6. WBV

Vibration therapy was performed on a vibration platform (Vibeplate 2424, Malcolm, NE, USA). Various body weight exercises of the lower- and upper-limb can be performed on a whole-body vibration (WBV) platform. Previous research has reported positive results in using upper-body WBV protocols to increase muscular force and power [40,41,42,43]. However, the use of upper-body WBV to expedite muscle damage is limited and warrants further investigation. The vibration was focused on the upper extremity by kneeling on a soft surface and then holding onto the outside edge of the platform in a partial push-up position during vibration. For this position, the elbows were partially flexed to damp vibration transmission to the head and to keep the elbow flexors more relaxed to mimic the resonant frequency testing condition (Figure 2). Participants were instructed to go half way down (elbows bent to 90°) and then come back up partly to hold an approximate 45–60 degree elbow flexion angle.

#### 2.6.1. Calculating Vibration Resonance of the Biceps Brachii

Currently, there is no known vibration frequency prescription for muscle recovery. Therefore, we selected a vibration frequency that closely matched the resonance frequency of the bicep brachii. The resonance frequency was determined by using the average bicep resonance frequency of a separate but similar age group of participants in a pilot study. The bicep muscle was excited using a pseudorandom signal and a vibration shaker to determine the first resonance frequency. The muscle response was measured using a Polytec PSV-500-3D scanning laser Doppler vibrometer (SLDV) (Polytec, Inc., Irvine, CA, USA). The Polytec laser system tracked the laser movement on the skin of the participant to determine the displacement, velocity and acceleration of the movement with respect to time. The recorded data was converted to displacement, velocity, and acceleration versus frequency. Each of the frequencies of these plots was correlated to different magnitudes of displacement, velocity, and acceleration. The first significant frequency peak was considered the first mode of the mass. This frequency caused the greatest displacement of the mass (i.e., muscle). This frequency was selected due to its ability to displace the muscle the greatest and therefore have the most significant vibrational effect on the participant’s muscle. The first resonance of the bicep brachii was measured between 15 to 18 Hz for all participants. Therefore, the average of the resonance range (16 Hz) was used as the vibration frequency in this study.

Vibrating an object at its resonance frequency elicits a larger harmonic amplitude response. It was assumed that exciting the muscle close to its resonance frequency would result in a greater harmonic movement response of the muscle. Prior research by Cochrane et al. [44] reported that low-frequency WBV resulted in small muscle length changes and increased muscle activation. Furthermore, muscle is reported to exhibit mechanical resonant-like behavior where maximum EMG response was found to be related to the resonance frequency of the muscle [45].

#### 2.6.2. Whole-Body Vibration Platform Assessment

Adhering to the WBV reporting guidelines [46], the input vibration from the Vibeplate was measured using a triaxial miniature accelerometer (356B1, PCB Piezotronics, Depew NY, USA, 14043). The accelerometer was glued to the Vibeplate next to the hand. When loaded by the person being tested, the Vibeplate induced an acceleration in the x-direction of x = 0.35 g (0.32 mm), the y-direction of y = 0.71 g (0.65 mm), and the upward z-direction of z = 0.11 g (0.10 mm). The x, y, and z axes are defined in Figure 3. The accelerometer data were sampled at 128K samples/sec for 5 s to ensure a frequency resolution of 0.2 Hz. Fifteen complex averages in the frequency domain were used to attain a steady state measure of the vibration. The standard deviation in the accelerometer measurements was less than 1%.

### 2.7. Statistical Analysis

The analysis for each dependent variable, VAS, MVIC, and SWE was a linear mixed models analysis blocking on participant. The explanatory variables of interest were treatment and time. The initial analyses also included age, height, and weight. These variables were nonsignificant and were eliminated from the final model. Post-hoc Tukey adjusted pairwise differences were computed for the interaction of treatment and time. This interaction was significant for each of the dependent variables. Histograms for VAS, SWE, and the percent change of MVIC did not reveal any gross outliers and that the distributions were normal enough to justify the use of a mixed models analysis. The VAS distribution was somewhat skewed. However, this was due to the number of zeroes in the control group, which are to be expected. All analyses were completed using SAS, version 9.4 (SAS Institute, Inc., Cary, NC, USA) with a level of significance of *p* < 0.01 with a pseudo-Bonferroni correction was utilized for multiple comparisons. Effect size (η^2^) was calculated to appropriately account for multiple comparisons. Minimal detectable change (MDC) scores were adjusted for the Bonferroni correction, and a Pearson Correlation Coefficient for MVIC and SWE was calculated.

## 3. Results

### 3.1. VAS

Full model analysis showed height as the only significant covariate (*p* = 0.0241). However, in the final reduced model, height was no longer significant (*p* = 0.0590) and demonstrated a low effect, accounting for only 1.8% of the variance (See Table 1). No significance was found for age (*p* = 0.1685) or weight (*p* = 0.1885). There was no significant difference in baseline VAS scores between groups after adjusting for height (*p* = 1.000). No significant differences were found between any time points for the control group (*p* = 1.000 for all). Both the V and NV groups demonstrated significant increases in VAS scores at 24 h and 48 h (*p* < 0.0001 for all) compared to baseline. The V group had significantly lower VAS pain than NV at 24 h (*p* = 0.0051), but not at 48 h (*p* = 0.029). The MDC of V compared to NV for VAS was 1.83 cm with a 95% confidence interval (95 CI) of (0.17–3.81) at 24 h, signifying a clinically relevant effect (since the 95 CI did not cross 0). There were no significant differences between baseline and 1-week for either the V or NV group (*p* = 0.7974 and *p* = 0.6441, respectively), indicating that VAS pain ratings had returned to baseline levels in one week for both groups (Figure 4).

### 3.2. MVIC

All MVIC values were transformed into percent change from baseline measurement for analysis. Final model analysis utilized both age and height as significant covariates (*p* = 0.0006 and *p* = 0.0083 respectively); however, both demonstrated a low effect and accounted for only 0.9% and 0.5% of the variance (See Table 1). There were no significant differences across time within the control group (*p* = 0.4477), but there were significant differences in treatment and time for both V and NV groups (*p* < 0.0001). Post-hoc analysis for within group differences shows that MVIC in the NV group decreased strength from baseline to post-ex (*p* < 0.0001), there was no significant change in strength loss from post-ex to 24 h (*p* = 0.814), and from 24 h to 48 h (*p* = 0.2123), but strength significantly improved from 48 h to 1-week (*p* < 0.0001). Post-hoc analysis for within group differences for the V group show that MVIC significantly decreased from baseline to post-ex (*p* < 0.0001), significantly improved from post-ex to 24 h (*p* = 0.0002), did not change from 24 h to 48 h (*p* = 0.9707), and significantly improved from 48 h to 1-week (*p* < 0.0001).

Post hoc analysis between groups showed the treatment groups (V and NV) were significantly different from control at post-ex, 24 h, and 48 h (*p* < 0.0001 for all). Only the NV group was significantly different from control at 1-week (*p* = 0.0016), but more importantly, still demonstrated significant strength loss compared to its own baseline (*p* < 0.0001). There were significant differences between NV and V groups at 24 h and 1-week (*p* = 0.0017 and *p* = 0.0008, respectively), showing that the V group initially accelerated the return of MVIC and fully regained isometric strength (+1.3%) at 1-week while the NV group exhibited a 10% decrement to baseline (Figure 5). The MDC for comparison of V to NV was 8.77% and calculation of 95 CI was (2.03–19.57) at 24 h and (2.85–20.39) at 1-week, demonstrating a clinically important difference since CIs did not cross 0.

### 3.3. SWE (Stiffness)

Full model analysis showed height as the only significant covariate (*p* = 0.0308). The final model analysis included height, which then became insignificant and demonstrated a low effect of accounting for 2.6% of the variation. There was a significant difference in the main effects of treatment, time, and treatment x time when controlling for height (*p* < 0.0001 for all). Post-hoc analyses revealed there was no significant change over time within the control group in SWE measurements. Both the V and NV groups displayed significant increases in SWE from baseline (*p* < 0.0001 for both). There was a significant decrease in SWE for the V group from 24–48 h (*p* = 0.0003), while there was no significant change in the NV group (*p* = 0.9341). Both V and NV groups showed a significant decrease in SWE from 48 h to 1-week (*p* < 0.0001 for both). There was no significant difference between V and NV at 1-week (*p* = 0.0214); however, from baseline to 1-week there was no significant difference in the V group (*p* = 0.7875), while there was still a significant difference in the NV group (*p* = 0.0015) (Figure 6). The MDC for within group change in SWE for both V and NV is 8.37 kpa with the 95 CI of baseline vs. 1-week at (−2.02–−18.75) for NV and (−7.51–9.22) for V, indicating a clinically important lack of change within the NV group since the CI did not cross 0, and no clinically relevant difference in the V group.

A Pearson Correlation between SWE and MVIC was calculated to be 0.72 with a *p*-value of <0.0001.

## 4. Discussion

To our knowledge, this is the first study to examine the efficacy of resonance vibration frequency of a WBV platform to improve muscle recovery following EIMD of the biceps brachii.

Current results show that UBV treatment led to a reduction in isometric force loss 24 h after the EIMD protocol, with a full return to baseline isometric strength at 1-week compared to the NV group. The MDC and 95 CI support the clinical importance of this finding at 24 h. This unique finding is contrary to most other WBV platform studies, reporting no differences in measures of isometric strength [16,47,48], concentric dynamic strength [16], or muscular power [23]. Other forms of localized vibration studies have also been unsuccessful in reporting improvements in isometric strength [49,50], concentric strength [49], and muscular power [51]. This lack of change in strength recovery occurs despite reported improvements in other measures such as pain perception, creatine kinase, and range of motion from the aforementioned studies. Conversely, an earlier study reported significant decrements in strength as a result of WBV treatment in the first 24 h following EIMD to the quadriceps muscles [52]. This, however, may have been a result of continued contractile work stress and large motor unit recruitment following an intense EIMD protocol.

Interestingly, only three vibration intervention studies to date have reported reduced changes in muscle strength loss after EIMD as compared to control conditions. Aminian–Far et al. reported knee extensor isometric torque loss to be attenuated with just one minute of WBV used prior to the EIMD protocol [17]. In another study, Bakhtiary et al. used one minute of localized vibration of the quadriceps prior to an eccentric protocol of decline treadmill walking and reported less isometric strength loss at 24 h compared to the no vibration control [53]. Both studies utilized WBV prior to their respective EIMD protocols, which appear to be less intense than many other EIMD protocols for similar muscle groups. Following an EIMD protocol in the biceps brachii, investigators observed a significantly reduced isometric force loss using an intervention known as whole-body periodic acceleration (pGz) [54]. Rather than transmitting vibration from a platform through the feet to the body, with pGz the whole body is passive and in contact with the platform and undergoes a constant low frequency/low amplitude acceleration for significantly longer periods of time. This unique form of intervention has reported cardioprotective circulation effects [55] and is postulated to add small pulses to the circulation, which increases pulsatile stress to the endothelium and enhances the production of nitric oxide (NO) [56]. Previously, it has been theorized that the antioxidant, anti-inflammatory, and antinociceptive properties of endothelial-derived NO could accelerate recovery from DOMS [54]. To date, very little has been reported on how NO may be influenced by WBV [57] or localized vibration [58], with both studies reporting contradictory results in diabetics. Currently, it is known that WBV enhances blood flow, and lower frequencies in the 5–25 Hz range have a greater effect on increased peripheral blood flow [59]. The use of both a lower frequency and matching resonance may have positively affected blood flow circulatory effects in our study. Future WBV studies should attempt to better determine circulatory effects in healthy and compromised populations.

WBV has been shown to facilitate muscle activation (to a greater extent during submaximal contractions [60]) and has been recognized as a form of “passive” exercise. Working muscle in turn increases metabolic demands and vibration through the lower extremities has been reported to increase blood flow [18], reduce arterial stiffness [61], and improve oxygen uptake [62]. As such, WBV exercise has also been shown to enhance microvascular blood flow and is hypothesized to function as an exercise mimetic with the potential to improve postprandial-related microvascular responsiveness [19]. As an exercise mimetic, the amount of muscle activation and resulting blood flow and oxygen uptake from metabolic demand is noted to increase with rising frequency and amplitude in WBV exercise [60,62]. However, the responses have been blunted compared to the exercise-specific activation of the same muscle groups [19]. It is plausible that lower WBV platform frequency used in the current study may have caused some blood flow alterations while minimizing muscle activation in the targeted biceps brachii. The present semi push-up position may have caused the biceps brachii to function as an antagonist, thus being in a more relaxed state for the vibration treatment. Therefore, a more passive targeted muscle group at a lower resonant frequency would be more similar to pGz (although its frequency is still higher) but may explain our strength differences between the groups. WBV based studies at lower frequencies targeting nonactive muscle groups are required to better observe the effects of vibration without compounded contraction and secondary exercise-related effects.

Our VAS pain scores suggest that vibration helped reduce pain perception more in the V group at 24 h, which is a clinically important difference as supported by the MDC and 95 CI values. While not statistically different at 48 h, VAS results demonstrated clinically meaningful changes in pain scores at both 24 and 48 h after the EIMD protocol, with both groups returning to baseline levels by 1-week. Previous research has reported that the minimum clinically significant difference in VAS pain scores is 9 mm, regardless of age, gender, and cause of pain [63]. The mean difference between pain scale ratings between VAS scores for the V and NV groups at 24 h was 2.0 cm and 1.5 cm at 48 h, which are larger than the 9 mm clinical reference. These findings are supported by earlier studies on DOMS where WBV was either reported to reduce DOMS pain via VAS more than the control [16,17] or equal to an exercising control [23,24]. Alternatively, an earlier study reported no difference in DOMS pain with either an active squat or active squat with vibration following EIMD [22]. However, the study recruited women only and a very short exercise and exercise + vibration protocol of only 2 × 30-s work sets for 3 days were implemented, which may indicate a threshold effect. Therefore, future studies should consider time/dosage comparisons.

To date, pain relieving effects of vibration are most commonly attributed to the gate-control theory via the stimulation of somatosensory afferents; however, supraspinal mechanisms may be involved [60]. Recent studies have reported a decrease in gamma aminobutyric acid neuron firing activity [64] and an increase in dopamine neuron firing rates [65] in the ventral tegmental area of the brain in response to approximately 50 Hz. While these studies suggest vibration can modulate supraspinal central nervous system substrates, greater work is needed to better elucidate how they affect descending nociceptive influence.

While our present study did not show any statistically significant differences in SWE responses between the V and NV treatment groups, SWE for the NV group at 1-week was 15.3% greater than at baseline while V was 1.6%. Considering the large effect size of “day”, the MDC and 95 CI for comparing baseline to 1-week supports that the lack of change in the NV group was clinically significant while the V group difference was not clinically different. Thus, the V group essentially returned to baseline stiffness values. While the exact related mechanisms cannot be explained, as previously described, WBV can improve blood flow [18] and microcirculation [19]. This may have altered the cascade of inflammatory responses typically associated with eccentric muscle damage. Neutrophils and monocytes are mobilized into the circulation, and the improved circulation may help with their transmigration into muscle where they breakdown damaged tissue and help with muscle repair [7,66]. The measurement of SWE in a lengthened position may be sensitive to the subtle changes in intramuscular inflammation, which involves the extracellular matrix [7]. Our SWE findings show that stiffness levels increase significantly in the first 48 h after EIMD, which corroborates the findings of Lacourpaille et al. (who reported an increase in SWE at 48 h after eccentric damage in both the elbow flexors and knee extensors following both low and high load EIMD protocols [26]). Interestingly, they also reported a correlation between an increase in SWE 30-min after eccentric damage and a decrease in peak torque at 48 h and suggested that the 30-min measurement could estimate the amount of muscle damage that may occur.

While there are a large number of studies utilizing SWE to assess muscle or tendon stiffness in various populations and pathologies, to date there are limited studies reporting on the use of SWE in relation to DOMS [26,38,67,68]. Other forms of elastography (such as acoustic radiation force impulse elastography (ARFI) and strain elastography) have also assessed DOMS [69,70,71]. Using strain elastography, muscle stiffness in the elbow flexors peaked 1–2 days following eccentric damage [69], which is similar to our SWE results. The authors also reported that strain elastography results were similar to durometer changes and that stiffness gradually decreased during the 4 days. Tsuchiya et al. evaluated elbow flexor stiffness using SWE at three different angles in response to supplementation following eccentric exercise [67]. The authors did not observe SWE changes in the 70° angle for the biceps, but they did observe increases over time with the elbow at 120° and 150°, which indirectly supports our measurement of the biceps brachii in elbow extension. Future studies tracking DOMS with SWE should consider measurement in a lengthened position as emerging data points toward increased reliability compared to resting positions [72].

The use of SWE may offer a unique option for tracking intrinsic muscle characteristic differences that may correlate well with both training and muscle stress/damage. When utilizing both magnetic resonance imaging (MRI) and SWE for DOMS over a 1-week period, shear wave velocity was reported to increase stiffness in the brachialis at 24 h, but did not correlate with fractional anisotropy or apparent diffusion coefficient MRI techniques [38]. Inflammatory responses after exercise is reported to be different than infection or tissue injury based on previous MRI and ultrasound investigation [73], and provides credence to emerging data for SWE use in muscle tissue. Other recent research has suggested that SWE is effective in determining muscle stiffness differences as a function of both distance and recovery time [68] and tracking physiologic muscle changes in ultra-marathon distance runners [25]. Further, SWE stiffness characteristics of the vastus lateralis are reported to be different in sprinters and long distance runners [74], indicating that SWE may be sensitive to intrinsic mechanical differences in muscle as a function of different mechanical loading and training programs. Further analysis of our data showed a strong negative correlation between SWE and MVIC (PCC = −0.72), suggesting that SWE stiffness increases as strength loss increases. To our knowledge, this has not been reported previously, and further research on this correlation may determine if SWE might be a strong predictor of a muscle’s intrinsic state of damage in the time course of DOMS.

In discussing the strengths of the present study, it is important to acknowledge its limitations. The current results are confined to the sample population of recreationally active participants without recent upper body resistance training. There is scope for further research to investigate trained athletes and females. An SWE postexercise measure was not taken due to the limited time of data collection. In previous experiments, we have observed that stiffness can be variable postexercise, possibly due to the varying degrees of reactive hyperemia and peripheral fatigue. Further, the current measurements at 24 h, 48 h, and 1-week are more reflective of the DOMS process where pain and strength loss gradually increase over time. The vibration frequency used for this study was selected from resonant frequency assessments, and although the average of the resonance range was calculated at 16 Hz, individualizing each participant’s resonant frequency may more accurately reflect the muscle response to future study designs. The current stiffness results should be contextualized to the present methodology of using the area calculation of the elastogram on our particular ultrasound device. There are various SWE systems that have different processing and algorithms to quantify elasticity and there is limited research comparing reproducibility between systems and ROI calculations. However, we are confident that the present ROI placement method was performed at a high level with a reported a range of 0.5–1.5% variation in repeated measurements of a particular elastogram.

## 5. Conclusions

This study is unique in that it has demonstrated a decrease in MVIC strength loss with the use of resonant frequency vibration for 4 days following an EIMD protocol in the biceps brachii. Based on previous WBV studies, it was expected that the use of UBV via a WBV platform would elicit decreases in subjective pain ratings through the DOMS cycle, but the additional results for MVIC may offer a unique mechanism to alter the normal time course of DOMS based symptoms. The ability to decrease pain perception and augment strength returns would have significant clinical utility if replicable. Furthermore, we have shown that the use of SWE may be a useful technique to determining the state of muscle recovery following eccentric damage. While this result is limited to the biceps brachii, further studies are needed to determine if this correlation to MVIC can be repeated in other muscle groups. The results of this study also reflect the need to observe the effect of the use of lower vibration frequencies in the time course of muscle recovery.

## Figures and Tables

**Figure 1 ijerph-18-07853-f001:**
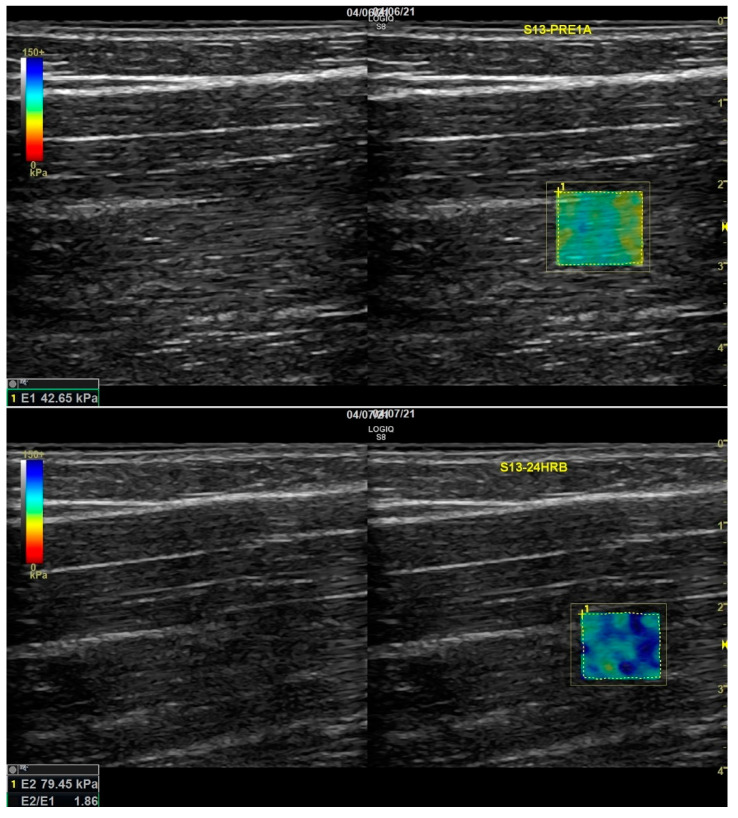
Example of shear wave elastography measurement at Pre (top = 42.65 kPa) and 24 h (below = 79.45 kPa) in an eccentric damage participant.

**Figure 2 ijerph-18-07853-f002:**
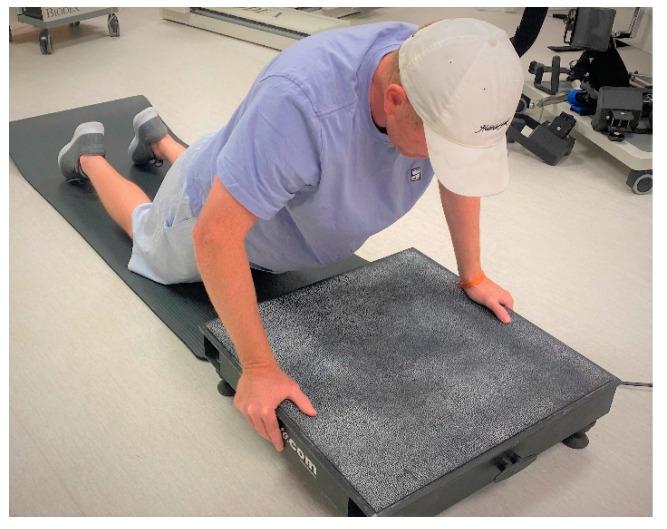
Semi push-up and hand positioning on the whole-body vibration platform.

**Figure 3 ijerph-18-07853-f003:**
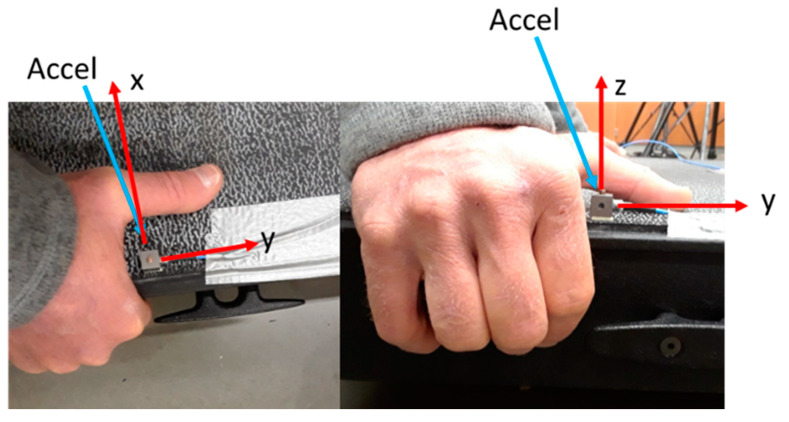
Triaxial accelerometer position on the Vibeplate used to measure the input vibration along all three axes (X, Y, and Z) while loaded with a participant’s body weight in the position used for vibration intervention.

**Figure 4 ijerph-18-07853-f004:**
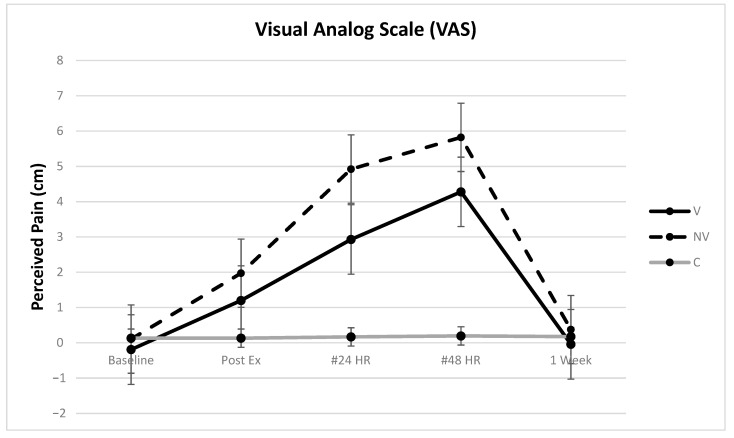
Perceived pain ratings using a visual analog scale. # = Control different from treatment groups (bars represent the 95% confidence interval values).

**Figure 5 ijerph-18-07853-f005:**
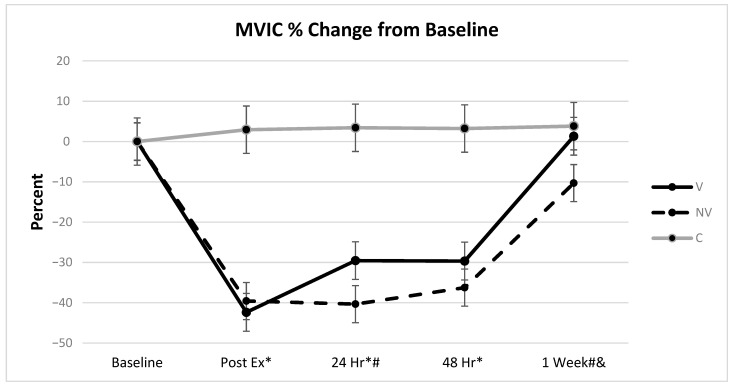
Change in maximal voluntary isometric strength. * = significantly different from control, # = V significantly different from NV, & = NV significantly different from control (bars represent the 95% confidence interval values).

**Figure 6 ijerph-18-07853-f006:**
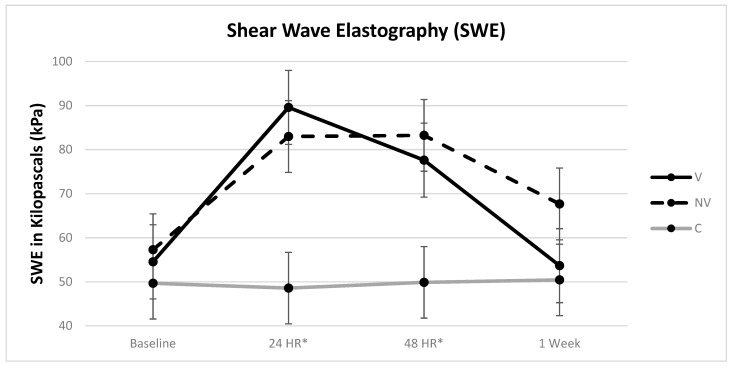
Shear-wave elastography values by group over time. * = control different from treatment groups (bars represent the 95% confidence interval values).

**Table 1 ijerph-18-07853-t001:** Significance and Eta2 of effects analyzed in the final model for each dependent variable. Minimal detectable change (MDC) scores are calculated for group comparisons for each dependent variable at the 0.01 significance level.

Dependent Variable	Effect	F-Value	*p*-Value	η^2^	Comparison	MDC
VAS	Height	5.23	0.0590	0.0177	C-NV	1.76 cm
	Treatment	16.19	<0.0001	0.1812	C-V	1.81 cm
	Day	42.09	<0.0001	0.3060	V-NV	1.83 cm
	Treatment * Day	11.41	<0.0001	0.1659		
MVIC	Age	12.67	0.0006	0.0090	C-NV	8.42%
	Height	7.30	0.0083	0.0052	C-V	8.60%
	Treatment	266.61	<0.0001	0.3798	V-NV	8.77%
	Day	69.73	<0.0001	0.3171		
	Treatment * Day	18.96	<0.0001	0.1984		
SWE	Height	2.95	0.0898	0.0258	C-NV	14.95 kPa
	Treatment	11.97	0.0002	0.2839	C-V	15.83 kPa
	Day	57.92	<0.0001	0.1803	V-NV	15.76 kPa
	Treatment * Day	20.17	<0.0001	0.1256		

## Data Availability

Data available upon request due to restrictions, e.g., privacy or ethical.

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
