# Peer review of "Effect of Resonant Frequency Vibration on Delayed Onset Muscle Soreness and Resulting Stiffness as Measured by Shear-Wave Elastography"

_ijerph, 2021, doi:10.3390/ijerph18157853_

Round 1
Reviewer 1 Report
In this study, the authors used resonant frequency vibration to the upper body (biceps brachii) to measure pain, stiffness recovery, and isometric strength after eccentric damage. The authors hypothesized that upper body vibration would improve symptoms of exercise-induced muscle damage by reducing subjective ratings of pain, improving isometric strength, and decreasing muscle stiffness. Participants were divided into three groups, control (no exercise), NV (exercise and no vibration), and V (exercise and vibration). Participants completed a visual analog scale (VAS), maximum voluntary isometric contraction (MVIC), and shear wave elastography (SWE) before exercise, and at 24h, 48, and 1-week after exercise. The authors observed significant differences in all tests, however the study was limited to the biceps branchii, and other muscles should be tested in further studies. The authors concluded that negative correlation between SWE and MVIC may be a predictor of the biceps branchii state in the time course of recovery.
The article is well written; the methods are very well explained, as well as the results. Probably the authors should explain why the female population was under-represented in the study. Females and males may perform differently and “gender” could have been an interesting covariate.
The authors mention and discuss the strengths of the study; however they should also discuss the limitations.
Minor observations:
Line 182: “…we have observed less variation in SWE values using a using an elastogram…” eliminate “using a”
Line 241: “…Vibeplate was measured using a, triaxial miniature accelerometer…” eliminate the comma after “a”
Line 321: “There was no sig difference between V and NV at 1-week…” if “sig” means “significant” write the whole word.
Author Response
Reviewer 1 - thank you for your comments and suggestions
Comments and Suggestions for Authors
In this study, the authors used resonant frequency vibration to the upper body (biceps brachii) to measure pain, stiffness recovery, and isometric strength after eccentric damage. The authors hypothesized that upper body vibration would improve symptoms of exercise-induced muscle damage by reducing subjective ratings of pain, improving isometric strength, and decreasing muscle stiffness. Participants were divided into three groups, control (no exercise), NV (exercise and no vibration), and V (exercise and vibration). Participants completed a visual analog scale (VAS), maximum voluntary isometric contraction (MVIC), and shear wave elastography (SWE) before exercise, and at 24h, 48, and 1-week after exercise. The authors observed significant differences in all tests, however the study was limited to the biceps branchii, and other muscles should be tested in further studies. The authors concluded that negative correlation between SWE and MVIC may be a predictor of the biceps branchii state in the time course of recovery.
The article is well written; the methods are very well explained, as well as the results.
Probably the authors should explain why the female population was under-represented in the study. Females and males may perform differently and “gender” could have been an interesting covariate.
RESPONSE: We appreciate your comment and this was a point of discussion as we established our methodology prior to the study and as we performed our statistics. We opted not to address any “gender” issues due to the limited number of females in the study and the inability to adequately power any effect of gender statistically. Two of the 3 subjects who dropped out of the study were female, leaving us with only 2 females in the control group and just 1 female in the V treatment group. Aside from a severe lack of statistical power, we did not discourage females from being in the study due to the fact that DOMS related research has not clearly shown a sex difference. In fact, most of the data points to the variation in age and training status explaining more of the variation in seen in sex (gender) differences which has been discussed many times in DOMS related reviews. So we appreciate your comments and we plan to address this issue with larger numbers and equal gender representation in our evolution of studies. This gender difference is also important to assess for the utility of the use of shear wave elastography.
The authors mention and discuss the strengths of the study; however they should also discuss the limitations.
RESPONSE: Thank you for noticing this. A paragraph discussing limitations to this study was added at the end of the discussion section.
Minor observations:
Line 182: “…we have observed less variation in SWE values using a using an elastogram…” eliminate “using a”
RESPONSE: Completed
Line 241: “…Vibeplate was measured using a, triaxial miniature accelerometer…” eliminate the comma after “a”
RESPONSE: Completed , thank you
Line 321: “There was no sig difference between V and NV at 1-week…” if “sig” means “significant” write the whole word.
RESPONSE: Completed
Reviewer 2 Report
Comment to the Author
ijerph-1259247
Effect of Resonant Frequency Vibration on Delayed Onset Muscle Soreness and Resulting Stiffness as Measured by Shear-wave Elastography
I appreciate the time you devoted in developing this manuscript. Here are my comments and suggestions for you.
This study examined the Effect of resonant frequency vibration on delayed onset muscle soreness and Resulting Stiffness in healthy people. The manuscript showed that resonant Frequency Vibration effectively decreased MVIC decrement and reduced VAS pain ratings at 24 hours post eccentric damage of the upper limb to a greater extent compared without vibratory stimuli.
Although the study addressed an interesting question, the study was not designed appropriately for the authors' conclusion. I have some serious concerns on the data analysis and the results.
Abstract
- Please clarify what UBV stands for.
Introduction
- For the introduction, you should describe more what kind of serious problems delayed-onset-muscle-soreness (DOMS) gives people.
- It is unclear why you thought that using shear-wave elastography (SWE) to assess stiffness would be useful for the DOMS intervention strategy; please clarify in the introduction.
Materials and Methods
- Are the V group and NV group exercise groups? Or are they the treatment group?
- The control group generally performs the same task, but with a different intervention or no intervention. In this study, the control group did not perform EIMD and vibration. This is not comparable to the treatment group. Probably the NV group corresponds to the control group in this study.
- Please clarify the definition of muscle damage given in this study.
- Why couldn't SWE be measured post-Ex?
- The study design should be accurately detailed including study type and the followed guidelines according to STROBE criteria (please, see https://www.equator-network.org/) Ann Intern Med. 2007; 147(8):573-577. PMID: 17938396
- How many people were assigned to each group? Are they matched for age and size?
- What is the EIMD protocol based on? And was the body size the same in each group?
- Why did you use whole body vibration? Is the effect different from applying local vibration to the biceps?
- As shown in Figure 2, it is difficult to imagine that the participants are in a relaxed posture.
- The definition of the position for vibration therapy is unclear and needs to be clarified.
Results
- Add the effect size η2 to the results of the linear mixed models analysis.
- For Fig. 4, why are some of the VAS values less than 0? The VAS should be calculated from 0-100.
- For all results, it is not correct to compare the exercise group with the control group. Since the controls did not receive any exercise or vibration intervention, it is natural that they would differ from the exercise group.
- For all results, please specify the p-value of the interaction of treatment and time.
- Please provide results that demonstrate the validity of this SWE in this study.
- All results should be summarized in a single table for ease of reading and consideration for the reader.
- A Pearson correlation coefficient value is not in the results.
Discussion
- Why did the vibration stimuli decrease muscle stiffness?
Please discuss its mechanisms.
- The clinical significance of this study is unclear. Please clarify it.
- Please describe the reasons for the strong negative correlation between SWE and MVIC.
Is this shown in your results?
- Limitations should be clearly stated.
Author Response
Reviewer 2 - thank you for your comments and suggestions
I appreciate the time you devoted in developing this manuscript. Here are my comments and suggestions for you.
This study examined the Effect of resonant frequency vibration on delayed onset muscle soreness and Resulting Stiffness in healthy people. The manuscript showed that resonant Frequency Vibration effectively decreased MVIC decrement and reduced VAS pain ratings at 24 hours post eccentric damage of the upper limb to a greater extent compared without vibratory stimuli.
Although the study addressed an interesting question, the study was not designed appropriately for the authors' conclusion. I have some serious concerns on the data analysis and the results.
RESPONSE: This study was designed to track the progress of subjects over time. Repeated measures were performed on each subject. With this in mind a repeated measures mixed models ANCOVA is the appropriate way to analyze the data. The concerns with the presentation of the results and specific comments have been addressed below.
Abstract
- Please clarify what UBV stands for.
RESPONSE: this definition was added
Introduction
- For the introduction, you should describe more what kind of serious problems delayed-onset-muscle-soreness (DOMS) gives people.
RESPONSE: Good idea. We added a short description in the first paragraph of common and potential functional effects of DOMS to better lead into research that has looked to alleviate its symptoms.
- It is unclear why you thought that using shear-wave elastography (SWE) to assess stiffness would be useful for the DOMS intervention strategy; please clarify in the introduction.
RESPONSE: Thank you for the suggestion. We added a sentence at the start of the SWE introduction to better convey why we used this assessment. We also made SWE info into its own paragraph in the introduction. Just to be clear, we did not use this as an intervention, but rather as an assessment tool since this form of ultrasound is for visual assessment rather than therapeutic.
Materials and Methods
- Are the V group and NV group exercise groups? Or are they the treatment group?
RESPONSE: See the response to the following comment as they are similar
- The control group generally performs the same task, but with a different intervention or no intervention. In this study, the control group did not perform EIMD and vibration. This is not comparable to the treatment group. Probably the NV group corresponds to the control group in this study.
RESPONSE: We agree with Reviewer 3 that the study design was a randomized single-blinded trial to determine the effects of a vibration treatment on common clinical measures of eccentric muscle damage in three groups – 1. received vibration; 2. did not receive vibration; 3, received no treatment or exercise program to cause muscle damage. The use of a true control group in this case was to hopefully show less variation and consistency in the variables of interest. Which was the case. While we expected relatively stable VAS and MVIC values, we were not completely sure what we would see from SWE. This is a pretty common thing to do in applied treatment based studies from our experience in our field.
- Please clarify the definition of muscle damage given in this study.
RESPONSE: We added a clarification for this in first paragraph of the “materials and methods” section. For our study, the indirect marker of strength loss following eccentric contractions was the predominant defining characteristic of muscle damage. Strength loss is viewed in DOMS literature as the most important characteristic of muscle damage as changes in muscle strength influence both the magnitude and course of changes in other markers of exercise induced muscle damage.
- Why couldn't SWE be measured post-Ex?
RESPONSE: SWE can be measured post-exercise. We did not measure SWE post-exercise primarily for timing reasons in the data collection process. Furthermore, we have found in our lab that with shorter bouts of exercise, stiffness tends to be quite variable (possibly due to varying degrees of reactive hyperemia; and peripheral fatigue in longer bouts). Even single bouts of normal exercise (non-damage exercise) have resulted in decreased stiffness acutely (Dankel & Razzano, 2020, Ultrasound; 23(4): 473-480). Our measurements at 24 hrs, 48rs and 1 –week were intended to be more reflective of the DOMS process which gradually becomes significant for pain and strength loss.
- The study design should be accurately detailed including study type and the followed guidelines according to STROBE criteria (please, see https://www.equator-network.org/) Ann Intern Med. 2007; 147(8):573-577. PMID: 17938396
RESPONSE: A randomized single-blinded trial was implemented to determine the effects of a vibration treatment on common clinical measures of eccentric muscle damage in three groups – 1. received vibration; 2. did not received vibration; 3, received no treatment or eccentric exercise to elicit muscle damage. A novel method measured muscle stiffness pre- and post- muscle damage. The reporting of this study conforms to the STROBE (Strengthening the Reporting of Observational Studies in Epidemiology) guidelines (von Elm et al, Ann Intern Med. 2007; 147(8):573-577). We have added this reporting statement at the end of first paragraph under “materials and methods”
- How many people were assigned to each group? Are they matched for age and size?
RESPONSE: More details on numbers per group and average age, height and weight are now included in section 2.2 of the paper. The subjects were not matched for age and size. We are not aware of previous studies showing size to be a significant factor for EIMD protocols or in expected DOMS responses. However overall conditioning and level of resistance training does have an effect. That is why we utilized a population that exercised but had not been involved in specific resistance training for the upper body.
- What is the EIMD protocol based on? And was the body size the same in each group?
RESPONSE: This is a good question and important to understand. Our EIMD protocol was based on what is considered to be the most important indirect marker for muscle damage, which is muscle strength loss. Thus, the protocol for EIMD was self-controlled (or normalized) for size since the amount of weight or resistance used for the EIMD protocol was based on the individual’s own strength (MVIC) at baseline.
- Why did you use whole body vibration? Is the effect different from applying local vibration to the biceps?
RESPONSE: You ask a great question. We have added some context for using a WBV platform in section 2.6 (WBV). Sorry for the longer answer here to address your localized question. Mechanically, applied vibration is very different from WBV in a few key ways. 1) WBV vibration input travels through multiple levels (joint segments), maintains matching input frequency and basically creates vibration from inside to outside, while localized vibration is more out to in. 2) The transmission of WBV is along the long axis of the bone segments while localized vibration is multiaxial in nature and limited in vibration amplitude (and often at a higher frequency), but has a greater effect on the outer layer of tissue. And 3) WBV is often applied in functional weight bearing positions while localized vibration can be applied either weight bearing or non-weight bearing. Applying direct vibration may acquire a different neurophysiological response due to the vibration being directly applied to the tendon or muscle at a greater vibration frequency and smaller vibration amplitude. Currently there is little research to suggest there are physiological differences and additional research is required to determine if differences exist.
- As shown in Figure 2, it is difficult to imagine that the participants are in a relaxed posture.
RESPONSE: We used a whole body vibration platform because it has become so widely used for both upper body and lower body exercise (with lower body being more common) in both rehabilitation clinics, sports medicine clinics and athlete training. The semi-push up position offered a unique opportunity to be weight bearing, but to have the biceps in a more relaxed state due to it being an antagonist to the primary muscles working to hold this position (triceps and pecs). So you are right that participants were not in a truly relaxed posture, but rather a functional weightbearing posture. They assumed a partial pushup position (within the range of 45- 60 degrees elbow flexion) which they had to maintain for each 60-second vibration exposure.
- The definition of the position for vibration therapy is unclear and needs to be clarified.
RESPONSE: This information has been added to the description in section 2.6 (WBV)
Results
- Add the effect size η2 to the results of the linear mixed models analysis.
RESPONSE: I have added the eta-squared and p-values for the final statistical model effects used for analysis and commented on those in the descriptive results paragraph for the 3 main dependent variables. The eta-squared estimate of the proportion of the variability of the variable. Low = below .1, moderate = .1 - .2 and large = .3 or higher.
- For Fig. 4, why are some of the VAS values less than 0? The VAS should be calculated from 0-100.
RESPONSE: You are correct; the actual measurements cannot be less than 0. However, because we are fitting a model with other variables we can get predicted values that are less than 0. This is caused by adjusting for the other variables in the model. What is represented in the graph is the predicted values we got from our statistical model.
- For all results, it is not correct to compare the exercise group with the control group. Since the controls did not receive any exercise or vibration intervention, it is natural that they would differ from the exercise group.
RESPONSE: We appreciate your comment on this and understand your viewpoint. Our response on why we had a true control group was included in an earlier answer to one of your questions. Technically you can statistically compare anything. The data we have presented allows for comparing all results and the comparison between both exercise groups is clearly represented in the data. The graphs represent both adjusted means for each measurement time as well as bars representing the 95% confidence intervals.
- For all results, please specify the p-value of the interaction of treatment and time.
RESPONSE: We have added a table based on our final model in the results section which includes the primary interaction effect. We have commented on significant p values in the results paragraphs describing the graphs. To create a table for p-values for each day/time for each group and each variable would produce 3 separate tables at 13 rows each (1 row for labeling). We don’t believe this is necessary to understand what is currently represented in the data as it is. We are willing to do this if the editor feels it is necessary.
- Please provide results that demonstrate the validity of this SWE in this study.
RESPONSE: We are not clear why this comment falls under the results section, however, we have added some context to the validity of this measure as established by previous literature in the middle of section 2.5 where we discuss the procedures for SWE.
- All results should be summarized in a single table for ease of reading and consideration for the reader.
RESPONSE: A table of the analysis of the final model was added. The addition of all post-hoc comparisons would be too large and we feel would be more cumbersome to evaluate. In our field the trend for data as shown in the line graphs presents the values and the significant variables identified. A mention of the significant p-values has been made in the results paragraphs.
- A Pearson correlation coefficient value is not in the results.
RESPONSE: Thank you, we have now mentioned the addition of this as well as effect size and MDC in the analysis section.
Discussion
- Why did the vibration stimuli decrease muscle stiffness? Please discuss its mechanisms.
RESPONSE: We have added some theoretical context to this in the discussion of SWE findings.
- The clinical significance of this study is unclear. Please clarify it.
RESPONSE: A statement of clinical relevance was added in the conclusion.
- Please describe the reasons for the strong negative correlation between SWE and MVIC. Is this shown in your results?
RESPONSE: This has been added to the results. We had this in the discussion, but we have added a brief description of the graphs for why this is a negative correlation.
- Limitations should be clearly stated.
RESPONSE: Thank you for noticing this. A paragraph discussing limitations to this study was added at the end of the discussion section.
Reviewer 3 Report
The authors completed a randomized single-blinded trial to determine the effects of a vibration treatment on common clinical measures of eccentric muscle damage and a novel method to measure muscle stiffness when compared to a group who did not receive vibration and a true control group (no treatment or exercise program to cause muscle damage). The study is of interest; however, in the current form is not suitable for publication for the following reasons:
1) The overall justification is limited and no sound theory is presented to rationalize the incorporation of vibration treatment following a DOMS protocol. More detail can be found below.
2) The decision to not normalize certain variables may be troublesome considering a height difference was observed between groups. This at minimum should be acknowledged, if the authors do not feel normalization is not necessary.
3) The statistical analysis does not allow for clinical interpretation. The authors should consider calculating and interpretation effects sizes and minimum detectable change scores for the variables of interest.
Introduction: The introduction provides a broad overview of the literature specific to DOMS, WBV, and SWE; however, never really makes a connection between the 3.
Actionable Items:
1) Provide a rationale for investigating the treatment of DOMS with WBV other than previous literature is inconsistent. What theory drives the potential positive outcomes when using WBV in patients with DOMS.
2) The purpose of the 3rd paragraph is not clear and includes too many independent ideas. It appears that the primary aim is to emphasize the clinical/research utility of the SWE and how it will provided more important information than traditional measures of isometric force, VAS pain, CK levels, etc; however, this is not clear. Specifically, the transition from line 63 to 65 create confusion of why is SWE necessary.
Line 38: Reword the sentence specific to cold water immersion etc as it contradicts the previous
Methods: Please include randomization technique and allocation of group assignment.
Line 103: Was there criteria around age or BMI? How many individuals ended up in each group?
Line 109: Was a specific intensity used to describe whether a person was considered recreationally active
Line 117: I would list other examples as well (pharmaceuticals such as NSAIDs)
Line 161: Please report the inter-session reliability as well considering the test retest study design
Line 237: I appreciate the rationale and approach to select a frequency; however, the authors do not provide justification that greater harmonic movement of a muscle equates to improved clinical features of muscle soreness following the exercise protocol. Where does the chosen frequency fall on the spectrum of targeting a-alpha, beta or even delta sensor fibers that most common pain control modalities target?
Line 268: The authors should consider including measures of magnitude of change such as Cohen’s d or Hedge’s g effect sizes with the respective 95% confidence interval. To further translate the clinical implications of the results, the authors could calculate minimum detectable change scores for each dependent variable, especially considering the novelty of the SWE variable. Then the authors can determine whether group differences exceeded the MDC (true change) or whether any changes were a product of the inherent variability of each measure.
Line 272: Please include direction language when applicable (V group had significantly lower VAS pain than NV at 24 hour). Also, please include means/standard deviations or mean differences. How much of a difference is really what is important, not just the statistical p-value. Also, include the additional analyses (effect size and MDCs).
Line 276: How do the authors think this would affect all other measures? Should everything be normalized to height or in the case of MVIC should the authors calculate torque?
Figures: What do the bars represent?
Author Response
Reviewer 3 - thank you for your comments and suggestions
The authors completed a randomized single-blinded trial to determine the effects of a vibration treatment on common clinical measures of eccentric muscle damage and a novel method to measure muscle stiffness when compared to a group who did not receive vibration and a true control group (no treatment or exercise program to cause muscle damage). The study is of interest; however, in the current form is not suitable for publication for the following reasons:
1) The overall justification is limited and no sound theory is presented to rationalize the incorporation of vibration treatment following a DOMS protocol. More detail can be found below.
2) The decision to not normalize certain variables may be troublesome considering a height difference was observed between groups. This at minimum should be acknowledged, if the authors do not feel normalization is not necessary.
RESPONSE: We appreciate the suggestion to add assessment of effect size (eta-squared) and have added the eta-squared effect sizes to a table showing the final model analysis for each of the 3 dependent variables. The overall effect of either height, weight, or age (if used for analysis) was very low and thus a form of normalization is not needed in our opinion and the opinion of our statistician.
3) The statistical analysis does not allow for clinical interpretation. The authors should consider calculating and interpretation effects sizes and minimum detectable change scores for the variables of interest.
RESPONSE: Effect sizes and MDC’s have been added to a table in the results section. Keep in mind that we presented the most conservative estimate possible for MDC which was based on the .01 significance level from post-hoc multiple comparisons. The MDC would be lower for each primary dependent variable if we had utilized the statistics of only the final primary model used for analysis.
Introduction: The introduction provides a broad overview of the literature specific to DOMS, WBV, and SWE; however, never really makes a connection between the 3.
RESPONSE: Thank you for this suggestion. We have revised much of the introduction to attempt to convey the need for each and feel the resulting purpose of the study helps tie this together. Any other suggestions to meet what you feel is lacking would be appreciated.
Actionable Items:
1) Provide a rationale for investigating the treatment of DOMS with WBV other than previous literature is inconsistent. What theory drives the potential positive outcomes when using WBV in patients with DOMS.
RESPONSE: In the revision of the introduction we have added some of the key physiological findings relating to a decrease in pain perception, improved blood flow and muscle temperature to suggest it is a modality that warrants investigation.
2) The purpose of the 3rd paragraph is not clear and includes too many independent ideas. It appears that the primary aim is to emphasize the clinical/research utility of the SWE and how it will provided more important information than traditional measures of isometric force, VAS pain, CK levels, etc; however, this is not clear. Specifically, the transition from line 63 to 65 create confusion of why is SWE necessary.
RESPONSE: Again, thank you for the suggestion. We feel that the revision of the entire introduction should help this to flow better.
Line 38: Reword the sentence specific to cold water immersion etc as it contradicts the previous
RESPONSE: Thank you, the two sentences were combined and should be more clear.
Methods: Please include randomization technique and allocation of group assignment.
RESPONSE: the randomization method of having the subjects draw a number from a bowl was added.
Line 103: Was there criteria around age or BMI? How many individuals ended up in each group?
RESPONSE: There was no criteria for BMI, height or weight. The only qualifying criteria is listed in section 2.2 (Participants section). For clarification, we added the age criteria of 18 years or older and we also added the breakdown of participants in each group with mean and SD for age, height and weight.
Line 109: Was a specific intensity used to describe whether a person was considered recreationally active
RESPONSE: No, intensity was not a factor. Our primary concern was avoidance of participants doing specific arm based resistance exercise as it would have a significant effect on their eccentric exercise bout response and subsequent DOMS (this is something we want to look at with future studies as elastography has been shown to vary depending on the mechanical stress placed on the muscle with various training types in running subjects). Our subjects met the listed criteria which was exercising at least 3 x per week for at least 30 minutes per session and the majority of these were generalized exercisers attempting to get cardio (walk, hike, jog, basketball, etc). Since the EIMD protocol included a significant number of reps (100), we wanted to avoid sedentary individuals due to risk of extreme responses (significant swelling or possible rhabdomyolosis) to the EIMD protocol.
Line 117: I would list other examples as well (pharmaceuticals such as NSAIDs)
RESPONSE: Yes, thank you for the suggestion. We originally had this in there in an earlier draft. We have re-added avoidance of both ice, topical ointments and medications.
Line 161: Please report the inter-session reliability as well considering the test retest study design
RESPONSE: Our reporting of ICC for the bicep measurement was based on pilot data prior to the study to help us determine our methods and final decision to measure at the extended position. We did repeated measures on 10 subjects at the same time each day for a period of 3 days to determine the ICC. Once we calculated the ICC’s we did not keep the data since it was not part of the actual study. We did not calculate intersession (or day to day) reliability at the time. We simply used that info to guide us in our methodology and we have done that with other muscle groups we are currently studying. As such we do not feel this is appropriate to expand on in the current paper.
Line 237: I appreciate the rationale and approach to select a frequency; however, the authors do not provide justification that greater harmonic movement of a muscle equates to improved clinical features of muscle soreness following the exercise protocol. Where does the chosen frequency fall on the spectrum of targeting a-alpha, beta or even delta sensor fibers that most common pain control modalities target?
RESPONSE: That’s a great question and to date most info on sensory fiber activation via vibration is theoretical. Vibration at a harmonic resonance can produce greater amount of movement… greater movement would move more tissue which would then be more likely to stimulate more sensory receptors thus increasing the sensory feedback into the spinal cord. Harmonic resonance will also produce vibrations on a broader frequency spectrum which could include more threshold activation of different receptors. Thus more muscle movement could occur at harmonic resonance with a decreased source amplitude which could make the overall acceleration and dosage of vibration lower… which would make the vibration exposure safer.
Line 268: The authors should consider including measures of magnitude of change such as Cohen’s d or Hedge’s g effect sizes with the respective 95% confidence interval. To further translate the clinical implications of the results, the authors could calculate minimum detectable change scores for each dependent variable, especially considering the novelty of the SWE variable. Then the authors can determine whether group differences exceeded the MDC (true change) or whether any changes were a product of the inherent variability of each measure.
RESPONSE: We have added both eta-squared effect sizes and MDC’s for interpretation.
Line 272: Please include direction language when applicable (V group had significantly lower VAS pain than NV at 24 hour). Also, please include means/standard deviations or mean differences. How much of a difference is really what is important, not just the statistical p-value. Also, include the additional analyses (effect size and MDCs).
RESPONSE: We have altered language direction where appropriate for all variables.
Line 276: How do the authors think this would affect all other measures? Should everything be normalized to height or in the case of MVIC should the authors calculate torque?
RESPONSE: Thank you for commenting on this. We have clarified the statistical results for either age or height in the context of their effect size when discussing their effects on SWE, MVIC and VAS. The calculation of eta squared showed small effects for the most part of these covariates (ranging from 0.5% - 2.6% of the variation). In the case of VAS we would not expect body morphology to have a significant effect in a normal healthy population. And for MVIC we would expect that after adjusting weight amounts for the eccentric exercise protocol based on individual strength, that body characteristics would normalize and have minimal effect on statistical results.
Figures: What do the bars represent?
RESPONSE: Thank you for catching that we did not put this in the legend. The bars represent the 95% confidence interval values.
Round 2
Reviewer 2 Report
Thank you for the resubmission.
You have responded very carefully.
However, I found some problems in the study design, experimental methods, and recruitment (age and body size matching). I hope that these will be improved in future studies.

Author Response
Thank you for your review. We completely agree that Study design, experimental methods, and recruitment (age and body size matching) is something that can always be improved upon when appropriate for particular measures in exercise science.
Reviewer 3 Report
I thank the authors for addressing some of my previous concerns and expanding their statistical analyses to include effect sizes and minimal detectable changes. However, I have a few remaining/new concerns:
1) The authors did not provide detail of how minimal detectable change scores were calculated
2) The authors elected to use an eta^2 effect size rather than Cohen's d or Hedge's g. The eta^2 effect sizes add value; however, an effect of magnitude (Cohen's d or Hedge's g) would assist with clinical interpretation of the results, especially if 95% confidence intervals are included.
3) Neither the effect size or MDCs are incorporated into the interpretation of the results. I encourage the authors to rely more on these analyses rather than the p-value when making recommendations considering the clinical nature of the outcome measures. Pre to post changes in these measures that do not have a moderate to large effect size with 95% confidence intervals that do not cross 0 have minimal clinical importance. Or, differences that do not exceed the MDCs may also have no clinical importance given that the observed changes may simply be from the inherent error of the outcome measure
Author Response
Thank you for your review and insight, it has improved our presentation. Based on your 3 primary comments regarding calculation of effects sizes, MDC and addition of 95 CI values we offer a response that we feel covers all 3 (since they are all integrated).
Our use of eta^2 is due to the fact that this most appropriately accounts for multiple groups and multiple comparisons. Cohens d and hedges g are both related to doing a two-sampled t test and doesn’t properly adjust for other variables. Thus eta^2 is a more complicated version of cohens d for going beyond a two sample t-test and adjusts appropriately.
The calculation of MDC values was not off the primary model. We took the pooled sample standard error the differences that we get in our model and multiplied them by a bonferroni adjusted t-value which adjusts for multiple comparisons.. which again more properly adjusts for multiple comparisons (as compared to the standard MDC formula which is based off of a 2-sampled comparison).... thus our results are inherently more appropriate and conservative for the analysis that was performed.
We added 95 CI values in the results for those comparisons that were statistically significant that we thought were particularly relevant in this study and referred briefly to those in our discussion of those primary variables.